# Characterisation of the Enzymatically Extracted Oat Protein Concentrate after Defatting and Its Applicability for Wet Extrusion

**DOI:** 10.3390/foods12122333

**Published:** 2023-06-10

**Authors:** Darius Sargautis, Tatjana Kince, Ilze Gramatina

**Affiliations:** Department of Food Technologies, Latvia University of Life Sciences and Technologies, LV-3004 Jelgava, Latvia

**Keywords:** oat protein, protein concentrates, functional properties, extrusion

## Abstract

An oat protein concentrate (OC1) was isolated from oat flour through starch enzymatic hydrolysis, by subsequent defatting by ethanol and supercritical fluid extraction (SFE) reaching protein concentrations of 78% and 77% by weight in dry matter, respectively. The protein characterisation and functional properties of the defatted oat protein concentrates were evaluated, compared and discussed. The solubility of defatted oat protein was minor in all ranges of measured pH (3–9), and foamability reached up to 27%. Further, an oat protein concentrate defatted by ethanol (ODE1) was extruded by a single screw extruder. The obtained extrudate was evaluated by scanning electron microscope (SEM), texture and colour analysers. The extrudate’s surface was well formed, smooth, and lacking a tendency to form a fibrillar structure. Textural analysis revealed a non-unform structure (fracturability 8.8–20.9 kg, hardness 26.3–44.1 kg) of the oat protein extrudate.

## 1. Introduction

To replicate the meat structure, alternative plant-based proteins are being successfully utilized as ingredients. Currently, the extrusion is considered as a general, well-recognized process for structurization and altering properties of plant-based proteins. The desired anisotropic structure of the product is achievable through a wet extrusion process [1,2]. Typically, soy has been considered as a major raw material for textured plant-based proteins to replicate meat analogues [3], along with wheat and pea proteins [4].

As an alternative to said raw materials oat protein might be of high interest due to its enhanced nutritional value and low toxicity [5]. However, only a few studies have been reported utilizing oat protein in texturized products. Some recent publications have been focused on oat protein extrusion, which was treated enzymatically by transglutaminase and protein glutaminase [4], or as a precondition for oil extraction [6], as well as in combination with pea protein [7,8], wherein high and low moisture extrusion technologies were applied, respectively. Moreover, the oat proteins that passed texturization have been reported as having a concentration that ranged from 14% [9] to 59% [10], or underwent texturization in mixtures with other ingredients such as wheat brans or potato compounds [11]. Such a high inclusion of supplementary compounds in a particular starch into compositions passing the extrusion process complicates understanding of protein behaviour, as itself, in extrusion systems.

In addition, the protein recovery methods influence functional properties of protein as a raw material. Oat protein might be recovered by a dry isolation method [12,13] as well as a wet isolation method, which subsequently assumes solvent extraction, precipitation or enzymatic treatment [14]. Most of conducted studies are dedicated to oat protein concentrates or isolates that passed alkaline extraction with subsequent protein precipitation shifting pH at an isoelectric point, thus later reporting functional properties of recovered protein [15] and their structure [16]. However, alkaline treatment typically might be considered as a process that initiates altering of the protein structure by formation of cross-linked amino acids and racemisation of L-amino acids into D-isomers [17]. Enzymatic extraction might be considered as an alternative protein concentration method with little impact on protein properties. Eliminating impurities such as starch and fibre could be achieved through treating material with enzymes excluding protein altering. A positive effect of hydrolysing starch and fibre was reported employing enzymes such α-amylases [18] and carbohydrases [19], respectively. Hence, the elimination of hydrolysed carbohydrates would allow us to increase the protein concentration substantially.

The presence of oil in oats is yet another aspect influencing protein texturization. A high content of oil prevents proper structure formation [20], though it might not be desired as an ingredient. Typically, oil is removed before protein extraction at the initial stage by mixtures of isomers, such as commercial hexane [21,22], which might be considered as an air pollutant and highly flammable solvent [23]. Oil extraction by ethanol could be an alternative solution; furthermore, ethanol is recognized in the healthy food industry. In addition, the oil extraction by supercritical fluids might be assumed as an optional possibility to reduce oil content in proteinaceous materials.

The aim of this study was to investigate an enzymatic oat protein extraction method, with the subsequent defatting of the obtained oat protein concentrate. The functional properties of oat protein concentrate were also the target of interest. In addition, the study examines the behaviour of oat proteins in an extrusion system along with its structure formation after passing extrusion.

## 2. Materials and Methods

### 2.1. Raw Materials

Commercial heat-treated oat flour from Helsinki mills (Vaasa, Finland) was used as a raw material, wherein max moisture content was 13% by weight. Particle size distribution by sieving analysis at 1000 µm, 300–1000 µm, and smaller than 300 µm was reported at the range of 0–1%, 0–5%, and 94–100%, respectively. Nutritional value was as follows: protein 10 g; carbohydrates 70 g of which sugar 0.4 g; fat 6.1 g, of which saturated fatty acids 1.1 g; dietary fibre 2.4 g, of which β-glucan 1.4 g. Enzymes, commercial, SQzyme HSAL, heat stable alpha amylase (HTAA) activities (from *Bacillus licheniformis*), and complex enzyme commercial enzyme Grainzyme FL with main xylanase and side beta glucanase (XBG) activities (from Trichoderma reesei) were kindly provided by Suntaq International, China. The referring source for the extrusion was a commercial soy protein concentrate produced by Shandong Yuxin bio-tech Co., Ltd. (Shandong, China) with the following composition: crude protein 70.6% (dry basis N × 6.25) by weight, moisture 6.7%, wt, ash 5.7% (dry basis) by weight, fat 0.7% by weight, particle size (100 mesh) 95% minimum. NaCl source was commercial grade table salt, ethanol, food grade 96.4°, water deionized, NaOH 0.1 M, HCl 0.1 M.

### 2.2. Chemical Characterization

Samples were characterized according the following methods: moisture ISO 6496:1999 [24], protein LVS EN ISO 20483:2014 [25], fibre ISO 5498:1981 [26], fat ISO 6492:1999 [27], amino acids LVS EN ISO 13903:2005 [28]

### 2.3. Oat Protein Extraction

Water was heated to a temperature of 60 °C, then HTAA and XBG enzymes were added at the range of 0.1% by volume each. The flour and enzymes ratio was based on earlier authors’ experiments. Room temperature oat flour was added at the ratio 1:10 by weight continuously stirring by Promix (Phillips, Hungary) hand mixer. The obtained mixture was periodically kept stirred (30 s for each 3 min) by Promix (Phillips, Hungary) hand mixer for 20 min at the temperature of 60 °C. Then, temperature was raised to 75 °C for 20 min, while stirring remained at the same interval rate, 30 s for each 3 min. The obtained hydrolysate was cooled down to 25 °C and passed separation by centrifuge Hereus Multifuge X3 (Thermo Fisher Scientific, Langenselbold, Germany), at the G-force 4400, for 4 min. The obtained biomass was washed by water at the ratio 1:10 and repeatedly passed separation, wherein separation parameters remained the same as previous. The washed biomass was dried in the hot air oven at the temperature 60 °C for 24 h. Dried protein concentrate was cooled down to room temperature and passed through the hammer mill LM 3100 Perten Instruments (Perkin Elmer, Waltham, MA, USA), sieve 0.8 mm. A milled sample of oat protein concentrate (OC1) was collected in a plastic sealed container and kept at room temperature for further analysis and processing.

### 2.4. Oil Extraction by Ethanol

OC1 was mixed with ethanol at the ratio 1:6 *w*/*v* continuously stirring by Promix (Phillips, Hungary) hand mixer for 30 s, then placed in sealed glass jars and kept in a hot air oven at 65 °C for 4 h. Time and temperature of the oil extraction using ethanol were based on earlier authors’ experiments. Glass jars were shaken by hand at the interval of about 5 s each hour. After 4 h, the supernatant was drained. The precipitate, solid biomass, was repeatedly mixed with ethanol at the ratio 1:3 *w*/*v* and stirred by Promix (Phillips, Hungary) hand mixer for 30 s and kept at 65 °C for 1 h in a hot air oven. The resulting supernatant was carefully drained and the precipitated biomass was dried in the hot air oven at the temperature of 65 °C for 24 h. The obtained oat protein concentrate defatted by ethanol (ODE1) was then naturally cooled down to room temperature, collected in a plastic sealed container and kept at room temperature.

### 2.5. Oil Extraction by Supercritical Carbon Dioxide

OC1 was defatted by applying laboratory scale SFE CO_2_ equipment (SFE 1000, Faneks, Ltd., Rīga, Latvia). The extraction chamber was filled with 130 g of oat protein concentrate. Extraction was performed with pure CO_2_ at the following conditions, flow rate of CO_2_—5.5 kg/h, extraction time 4.5 h, pressure in the range 285–300 bar, and temperature 50 °C. The mixture of yellowish dim oily components was collected in a separate vessel. The mass percentage of extracted oil was calculated via determining oil content in dry matter of treated material before and after CO_2_ extraction. The obtained sample of the oat protein concentrate defatted by SFC CO_2_ (ODC1) was collected in a plastic sealed bag and kept at room temperature.

### 2.6. Protein Solubility Index of Defatted Oat Protein Concentrate

The samples of ODE1 and ODC1 were subjected to solubility treatment in aqueous solutions wherein pH was set at 3, 5, 7 and 9. The method of nitrogen solubility index was carried out as described by Morr et al. [29] and Sewada et al. [30] with minor modification. Samples in the amount of 1 g were dispersed in 0.1 M NaCl solution. The dispersions were adjusted to specific pH values of 3, 5, 7 and 9 with 0.1 N HCl or 0.1 N NaOH to final volume of each dispersion 50 mL, then continuously stirred with a magnetic stirrer for 2 h at 25 °C. The values of pH were measured by means of a pH-meter Mettler Toledo Seven Compact equipped with an Inlab Expert Pro-ISM pH-electrode. The dispersions were then separated by centrifuge Hermle Z 206 A (Hermle Labortechnik GmbH, Wehingen, Germany) at G-force 4600 for 5 min (25 °C). The centrate was filtered through filtration paper (GOST 12026-76, FB-III-20, TU-2642-001-68085491-2011, ash content, no more than 0.00133%, filtration capacity < 26 s, bursting strength 5 kPa, Melior XXI, Ltd., Moscow Oblast, Russia). Nitrogen content in filtrates was determined by Kjeldahl method. Nitrogen solubility index was calculated according Equation (1):NSI = (Nitrogen in filtrate (%) × weight of solution (g))/(Nitrogen in dried sample,% × weight of sample (g)) × 100,(1)

Nitrogen to protein conversion factor was set at 6.25.

### 2.7. Water/Oil Holding Capacity

Oil and water binding capacity were determined as described by Mirmoghtadaie et al. [31] with slight modifications. One gram of sample was dispersed in 10 g of commercial sunflower refined deodorized cooking oil or ultrapure water and stirred by heavy duty vortex mixer VXHDDG (Ohaus, Parsippany, NJ, USA) for 1 min at 25 °C, where speed was set at 1200 rpm and 2500 rpm, for oil and water, respectively, and left for 30 min. During rest time, the samples were periodically shaken for 10 s each 10 min by heavy duty vortex mixer VXHDDG (Ohaus, Parsippany, NJ, USA). Then, the dispersions were separated by centrifuge Hermle Z 206 A (Hermle Labortechnik GmbH, Wehingen, Germany) at G-force 3000 for 5 min. The supernatant was poured and the pellet weighted. The oil and water holding capacities were determined via expressing the amount of water and oil in grams retained per gram of protein concentrate, respectively.

### 2.8. Foaming

One gram of oat protein concentrate was mixed with 33 g of ultrapure water at pH 7 in a 50 mL graduated glass cylinder then continuously mixed for 30 min with a magnetic stirrer. The dispersion was then continuously stirred with high shear mixer T10 Ultra Turrax (IKA Werke GmbH & Co. KG, Staufen im Breisgau, Germany) for 5 min. The total volume of foamed mixtures was recorded at 5, 10, 30, 60 and 120 min. The method was adapted with slight modification as described by Mirmoghtadaie et al. [31]. Foaming capacity was calculated according the Equation (2).
Foaming capacity = VF/Vi × 100 (%),(2)
where: VF is a foam volume in millilitres, and Vi is an initial volume of the aliquot in millilitres.

### 2.9. Molecular Weight Determination

The sample powders were solubilised using in 6M urea and 2% SDS buffer. All samples were diluted to 4 mg/mL (scales TE-124S-OCE, Sartorius AG, Gottingen, Germany), then shaken for 1 min by heavy duty vortex mixer VXHDDG (Ohaus, Parsippany, NJ, USA) at 2500 rpm. The suspensions were then shaken for 1.5 h by environmental shaker incubator ES-20 (Biosan, Ltd., Riga, Latvia) at room temperature. The suspensions passed 15 min centrifugation at G-force 2300 (centrifuge CM-6MT, Elmi Ltd., Riga, Latvia). The resulting supernatants of protein samples were collected and frozen.

Analyses were performed by sodium dodecyl sulfate-polyacrylamide gel electrophoresis (SDS-PAGE) under reducing conditions using Agilent Bioanalyzer 2100 capillary electrophoresis system (Agilent, Santa Clara, CA, USA) with Agilent Protein 230 Kit (14–230 kDa sizing range). Briefly, aliquots of 4 µL unfrozen protein samples were mixed with 2 µL DDT denaturing solution, prepared according Agilent protocols (3.5 Vol, -% of 1M DTT), spined for 15 s and then heated at 95 °C for 5 min, cooled down and diluted to 90 µL with deionized water. Ladder, Gel-Dye mix and destaining solution were prepared and loaded according the Agilent assay protein protocols for Bioanalyzer 2100.

### 2.10. Extrusion

#### 2.10.1. Raw Material Preparing

ODE1 was mixed with pure water to the final moisture of 55% by weight. The water was dispersed on the top of the ODE1 and mixed by spatula to ensure homogeneity of distribution of raw materials. The blended material was left for a half hour at room temperature in an open container before the extrusion process.

#### 2.10.2. Extrusion Process

The blend of the prepared raw material was processed in a single screw extruder Extrusiometer L20 (Göttfert, Buchen, Germany). The diameter of the installed screw was 20 mm, wherein the length diameter ratio was L = 25 D, coefficient of the compression 1:1. The temperatures for 3-barrel heating zones were set at 90–110–130°, which were automatically controlled by installed electric heaters and forced air cooling. The screw rotation speed was set at a constant speed at about 50 rpm to keep pressure in the range of 5–8 bar at the end of the barrel. The die providing a shear stress had a square 13 × 13 millimetre slit, and feed rate was not calculated. The raw material was continuously provided manually keeping the feed end always filled and pressed assuring what the raw material was sufficiently fed.

Some of the representative samples of the protein extrudates were sliced in tangential and cross-sections and passed through a freeze drier for moisture elimination. The remaining samples were dried in the freeze drier without size reduction at the form obtained in the extrusion process. Freeze drying was performed for about 48 h until moisture was removed in samples.

### 2.11. Textural Analysis

The textural properties of extrudates were analysed by TA-HD Plus texture analyser and data were generated by Exponent software (Stable Microsystems Ltd., Godalming, UK). For each sample of protein extrudate, at least 5 measurements were completed. The cutting parameters: pre-test speed 1 mm s^−1^; test speed 5 mm s^−1^; post test speed 10 mm s^−1^; cutting distance of 20 mm into the extrudate sample; and trigger force 0.049 N. The samples were cut into cross-sections using an HDP/BSK probe (standard blade set with knife). Generated values of peak force (kg), peak positive force (kg) and positive area (kg × s) were expressed as fracturability, hardness and toughness [32,33] respectively.

### 2.12. Colour Analysis

Colour measurements were performed for initial untreated raw materials and extrudates. The extrudates were collected and measured within 30 min after extrusion. Colour was determined using colorimeter Color Tec-PCM/PSM and software ColorSoft QCW (Accuracy Microsensors, Inc., Pittsford, NY, USA) evaluating colour in CIE L*a*b system. The total colour difference (ΔE) was calculated according to the equation Nr. 3 [34,35].
(3)$$\Delta E=\sqrt{{\left(L^{*}-L_{0}^{*}\right)}^{2}+{\left(a^{*}-a_{0}^{*}\right)}^{2}+{\left(b^{*}-b_{0}^{*}\right)}^{2}}$$
where: *L**—colour intensity (light-dark) in extrudate; $L_{0}^{*}$—colour intensity (light-dark) in the initial material; *a**—green-red colour component in the extrudate; $a_{0}^{*}$—green-red colour component in the initial material; *b**—blue-yellow colour component in the extrudate; and $b_{0}^{*}$—blue-yellow colour component in the initial material.

### 2.13. Scanning Electron Microscope

Structures of the surfaces of the dried extrudates were analysed by a scanning electron microscope (Tescan MIRA3 XMU) without any surface treatment required. The magnification, voltage, and the segment value in microns were automatically reported and seen in each micrograph below. The focus location was chosen from the point of view of the best representative visibility of the sample.

### 2.14. Statistical Analysis

Friedman rank sum test was applied via analysing median differences among polar and one way ANOVA for non-polar amino acid groups with prior Shapiro–Wilk normality test. T-test was applied for textural analysis. ANOVA tests followed by Tuckey’s HSD and compact letter display were applied for the remaining analyses unless stated otherwise. Statistical analysis was conducted in R [36]. Figures and data were processed using packages [37,38,39,40]. RStudio [41] was used for Integrated Development Environment for R.

## 3. Results

### 3.1. Protein Extraction

Protein extraction was performed in five batches, with an initial amount of flour of 2000 g. The amount of proteinaceous biomass in the underflow averaged 1039 g, while the second wash reduced the averaged amount to 703.25 g of the proteinaceous biomass, therein dry solids accounted for about 33%. The dried sample counted protein at 63.05% in dry matter. The protein yield was about 77.0%, comparing to the initial protein content in oat flour. Such a high recovery yield of protein could be attributed to the oat globulin properties, specifically high insolubility of oat protein in aqueous solutions at neutral pH, which is discussed in more detail below in 4.4. In addition, a high concentration of protein tends to form insoluble protein aggregates due its increased tendency for intermolecular bonding to surrounding proteins in their vicinity [16]. Yet another factor increasing protein recovery could be attributed to surface hydrophobicity of protein, which exposes negative correlation with protein solubility in aqueous solutions [42]. The results characterizing samples are presented in Table 1.

Worth mentioning is the fact is that oat protein was recovered from oat flour, wherein major parts of crop tissues not related to endosperm were removed prior during oat processing. That decreased the loss of yield, as most of the water soluble albumins are located in embryonic axis and scutellum [43], which typically undergo separation. Furthermore, the aleurone layer, which is a major location of enzymes [44,45], was also removed to some extent from oat flour during processing. That allows us to state that protein concentrate mostly consists of a globulin fraction only.

On the other hand, high protein recovery should be associated with protein redistribution at the initial milling process, and the substantial part of protein remains in fibre fraction, as separated crop tissues were reported as having the highest protein concentration [43].

### 3.2. Oil Extraction

The oil was extracted by treating oat protein concentrate via ethanol and supercritical fluid CO_2_. The results of the oil concentration in samples before and after defatting are presented in Table 1. The oat flour initially contained 6.2% of oil, although the concentration of oil in oat protein concentrate increased up to 20.6%. Considering the yield of protein recovery that resulted in a substantial amount of oil of about 43% of the total, which was concentrated in the solid part of protein recovery. Such a relatively high oil concentration could be attributed to the high content of non-polar oat lipids, which typically may vary in the range of about 65% to 90% of the total [46], which are typically not extractable with water without prior specific treatment. Defatting the oat protein concentrate by ethanol and supercritical fluid CO_2_ reduced crude oil content in the material from 20.6% to 4.9% and 3.5%, respectively. Treating proteinaceous materials by short chain alcohol solvents, including ethanol, assumes a higher extractability of phospholipids when compared to non-polar solvents, such as hexane [47]. Ethanol purity was also reported as a critical factor affecting oil extraction [48]. The presence of water in ethanol reduces lipids extractability, as well as the temperature [30]. It could be speculated that the remaining oil in the protein concentrate contained a higher ratio of nonpolar lipids than the initial material. Although, interestingly, higher extractability of oil in a supercritical system was observed despite the fact that CO_2_ is considered as a fluid which extracts presumably non-polar oil. Typically, some co-solvents might be considered to be included into the CO_2_ system to increase extractability, including ethanol [49].

### 3.3. Amino Acids

The amino acid composition of analysed protein in the initial raw material, extracted protein concentrate, defatted by ethanol and supercritical fluid and indispensable amino acid requirements in adults regarding FAO recommendations are illustrated in Table 2.

Tryptophan was not indicated along with other amino acids due to its decomposing at acid hydrolysis. The concentration of amino acids expressed per gram of protein in total reflects the protein redistribution through the recovering steps. Compared to other concentrations of amino acids, the abundance of glutamic acid is obvious, though this is a typical phenomenon with crops, including oats. The concentration of protein through starch hydrolysis noticeably increased the concentration of amino acids such as Tyr, Cys, Arg, Met, and Ile, each above 10%. A slight increase, less than 10%, was observed in the amino acids Val, Pro, Glu and Asp. The loss of the amino acid concentration was determined for Lys, Leu and Thr, which counted 9%, 9% and 6%, respectively. The decrease concentration of amino acids such as lysine and alanine might indicate those were related to albumins that might pass into the liquid phase during hydrolysis and separation, as concentrations of said amino acids in albumins were reported as having increased levels comparing to globulins. Interestingly, the noticeable increase concentration in Tyr could indicate the concentration of glutelins, which, respectively, were high in glutelin fraction [43].

Oil removal by supercritical CO_2_ fluid led to a slight decrease in Ile and Pro, yet concentrations of other amino have been increased. Ethanol extraction significantly increased concentrations of His, Val and Ala, which were determined as higher at 14%, 9% and 8%, respectively.

Sorted by their polarity, amino acids fall into two classes, polar and nonpolar. Amino acids classified as non-polar are free of hydrogen donor or acceptor atoms and those included Gly, Phe, Met, Leu, Ile, Pro, Ala and Val. Side chains of these amino acids are considered as being prone to form clusters and be buried inside the protein matrix. The remaining analysed amino acids were classified as polar, including charged amino acids, both positively and negatively charged [51,52,53,54]. Those considered as being hydrophilic and are usually found outside proteins [55]. Typically, oat albumins contain higher concentration rate of polar amino acids, yet increased amount of nonpolar amino acids are concentrated in oat globulin [42]. The polarity of the amino acids exhibited little influence on oil extraction method. The composition of amino acids did not change substantially, yet the concentration of amino acids increased in both cases, wherein the increased summarised amount was significant for all cases at *p* < 0.05, except for non-polar extracted by supercritical CO_2_ (see Figure 1). This fact could be assumed to indicate a decrease in the summarized amount of amino acids excluded from the report.

The values of essential amino acids, except Lys, surpass FAO requirements for essential amino acids. It is worth mentioning that the values of the recommended composition of essential amino acids are quite relative indicators. Other influencing factors should be considered such as the efficiency of the utilisation of amino acids, as those critically depend on total nitrogen in the diet. Higher total nitrogen intake assumes lower intake requirements of essential amino acids [50].

### 3.4. Protein Solubility

The oat protein concentrate solubility dependence on pH was presented in Figure 2. The oat protein was practically insoluble within the entire pH range tested. While the highest rate was slightly above 9% at pH 9, the solubility at neutral or close to this value was negligible, moving close to 6%, for both observed materials. The pH range was pre-determined assuming the applicable use of pH in food applications. Protein is susceptible to be hydrolysed at extreme pH to some extent [56], reducing molecular size, that correspondingly should increase protein solubility [57]. However, even at the extreme measured values, the solubility was significantly lower than expected. Similar results were obtained via enzymatically extracting oat protein from oat flakes with subsequent separation of suspended solids and washing [58]. Despite introducing other raw materials, such as oat flakes, the oat protein resulted in relatively similar solubility (see Figure 2). Those observations were rather controversial findings, as even at a critical pH of 9 at which the typically reported solubility reaches from 20% [59] to 70% [54] for protein extracted by alkaline solubilisation with consequent precipitation at the isoelectric point. Native oat proteins obtained by air classifications even at pH 7 were more than 20% soluble [60]. An interesting observation was revealed when oat protein concentrated enzymatically demonstrated lower solubility at pH 9, than at pH 5, which reached about 10% and about 50%, respectively [61].

It was also reported that the solubility of purified oat globulins had very limited solubilisation [62]. Although the introduction of salts provided very different results, the solubility was significantly increased up to 90% when the NaCl concentration reached up to 1 mol/L. On the other hand, oat globulins having salt soluble protein fraction behaviour significantly depends on the salt concentration, not only the pH value. Interestingly, a recently reported investigation revealed that low ionic strength could even reduce oat protein solubilisation due to its substantial increase in size at low ionic strength [63]. It is worth mentioning that the methodologies of the protein solubilization differ, though some limitations were reported [64]. The proposed standardized procedure for protein determination in food includes 0.1 M NaCl solution [29,30] apparently reducing values of protein solubility, which might reflect the current study as well.

### 3.5. Water and Oil Holding Capacity

The water and oil holding capacities of defatted oat protein concentrates are presented in Figure 3.

The results demonstrate that the defatting method has little influence on the ability of oat concentrates to bind water and oil at different capacities. In both cases, the results were in a similar range among the extraction methods, although the difference among the solvents is substantial. The water was found to be more able to bind to the oat protein concentrate than the oil in both cases. The results of oil and water binding capacity are close to oat protein concentrated by alkaline extraction method [15], although higher than in the protein concentrate, which was extracted from different oat fractions such as oat brans or whole oat flour. Those bind water at the range from 1.27 g water/g protein to 1.71 g water/g protein. It should be considered that the oat fractions prior to alkaline extraction were defatted by hexane. The presence of a beta glucan fraction was reported as significantly enhancing the water binding factor [59].

### 3.6. Foamability

Foaming properties might be described as including foam capacity and the foam stability specifics. While the foam capacity represents the total amount of foam produced under specific conditions, the foam stability might be attributed to depending on time the precipitation rate of liquid [65]. Determined foamability of the samples are presented in Figure 4. The sample of the protein concentrate defatted by a supercritical fluid system produced more foam than the sample treated by ethanol. The remaining lipids might influence the foaming capacity. Yet, oat protein isolate obtained through alkaline extraction (0.015 N NaOH), which underwent foaming initiating by introducing air from the bottom of the column through the fritted disk, gave a significant increase in foam, 20 mg of protein in the 20 millimetres gave foam more than 100 millimetre (measured in 250 mL glass column), yet after 15 min the foam decreased to 30% of the initial foam [66]. Relatively high foamability was observed in defatted by hexane oats, later passed to alkaline treatment with the following isoelectric precipitation. The foaming capacity of the obtained protein in all samples was higher than 100% and in all cases it exceeded the foaming capacity of non-defatted oat protein concentrates [67]. Interestingly, the reported foamability of oat protein concentrate, which was extracted from oat brans treating them by enzymes, even overcome the oat protein recovered by the alkaline method [61]. It is believed that one factor related to such a limited foaming capacity of the samples could be referred to the negligible solubility of the oat protein, as the protein solubility is one of the key factors determining foaming capacity [65,68]. The remaining lipids in oat protein concentrate could also be referred to as the limiting factor or foaming capacity, due to its weakening interaction between proteins adsorbed into oil or bridge forming between protein surfaces [69]. Surface activity should also be attributed to foaming capacity factors and it should be considered as a key property determining foam capacity through adsorption speed and the ability to decrease in the surface tension [70]. Protein susceptibility to hydrogen bonding, its hydrophobic interactions or covalent bonding play important roles during film formation [60]. The protein molecular size might also significantly influence the foaming capacity. It is believed that the large molecules might support proper foam formation due to the exposed large quantity of interacting points. Although, it has been recently reported that a smaller molecular size might in some cases increase the surface activity in the solute even to a higher level. It was demonstrated that soluble soy molecules, smaller in size, were more active at the surface [57]. We could conclude that the oat protein concentrate has a very limited foaming capacity.

### 3.7. Molecular Weight of Oat Protein Concentrate

The molecular weight of oat proteins was analysed to reveal whether the enzymatic extraction followed by defatting generates noticeable differences in oat protein analysed by SDS-Page in reduced samples. In the ODC1 sample, the bands with 25.4, 26.8, 42.5, and 48.3 kDa counted 16.8, 25.2, 20.4, and 27.7% of the total protein amount, while the bands of 26.3, 27.6, 43.8, and 49.8 kDa in sample ODE1 represented 23.3, 28.7, 20.8, and 26.0% of the total determined protein. The size dispersion among the samples was clearly noticeable within the two major areas at 46 kDa and 28 kDa, highlighted as A and B areas in Figure 5, respectively. The patterns of both samples were similar, bands among the samples lied in the similar range without substantial differences. This confirms that oat protein size was not impacted by the extraction method of the lipids. Some reports speculated that the electrophoretic profile of oat proteins might be influenced by the defatting treatment as itself [67]. The size of endosperm proteins bounded to oil bodies has also been reported, which was the most prominent at the 28 kDa band [71]. However, lipids act as a contaminant in electrophoresis, which are assumed to be removed to improve accuracy of measurement [72,73]. Size of 50 kDa band was reported as the main in oat protein concentrate at non-reducing conditions, although mercaptoethanol-reduced samples represented protein mostly at 20 kDa and 30 kDa bands, which were attributed to β and α protein subunits, respectively [7]. Reported data of fractioned protein revealed results comparable to the present study. Oat globulins extracted from oat brans produced bands with molecular weights of 15.7, 28.8, 38.8 and 42.7 kDa [42]. Finally, the published patterns of isolated oat globulins almost identically corresponded to data of the present research [74]. Such a high accordance in protein size assumed the composition of the investigated protein was comprised mostly of oat globulins.

### 3.8. Extrusion

Empiric observation of extrudate might be specified as a sample whose structure was firm, well-formed solid shape, dark pale brown coloured, and with visible rare cracks. During the extrusion, some vapour was noticeable, as the die was without cooling, the extruded material left extruder as free form at the temperature naturally occurred after decompression. The extrudate did not tend to expand, and the dimensions were kept close to slit dimensions, breaking in non-regular length pieces. Cooled down to room temperature, it become difficult to slice. Samples of oat protein extrudate’s cross and longitudinal sections are represented in Figure 6.

It was mentioned that the protein content should reach the range of 50% to 70% to ensure the fibrous structure could be formed during the extrusion [7]. However, other key factors might influence the formation of a fibrous structure in the oat proteinaceous material, such as pressure, cooking time, temperature or inclusion rate of water or other crop compounds, such as starch [1,75]. It was recently reported the determinants, including the temperature and moisture, influencing the formation of the fibrous structure. When parameters of one of the mentioned specifications were set at the constant level, while increasing the other specification, the formation of the fibrous structure was affected negatively [35]. The shear force and its direction were mentioned as being important specifications in fibrous structure formation, as well. It might be assumed, that the said specific determinants such as moisture’s content influences the shear force, as an indicator of materials’ ability to form fibrous structures. Another yet noticeable factor is the oil content in the initial material. As the substantial part of oil during the protein concentration typically remains on the protein side, its high (oil) content prevents proper or even excludes the process of the formation extrudates. One of the reasons preventing proper formation of extrudates is a reduced friction in the extrusion system. The authors’ experiments [75] revealed that oil content of 20% reduces extrudates’ ability to form solid structures, the product coming out from the die was in lose form, ready to break down, and without a visible molten protein matrix. The pressure applying the same die as mentioned could not be set at a constant level due to the lack of or reduced shear forces and has a dynamic state.

It is believed that the high protein as well as dietary fibre content reduce extrudates’ expansion [76]. High protein content also initiates clustering of the extrudate, as well as porosity and cell thickness. The formation of protein aggregation increases noticeably during the high moisture extrusion process [65], consequently reducing the solubility of extrudates and foaming capacity. It was also noted the increase in free sulfhydryl level, correspondingly severely affected disulphide bonds which influence structure formation. However, increase in moisture content decrease free sulfhydryl in extrudates. The lack of homogeneity and protein reorganisations into larger size aggregates was also observed during corn protein extrusion [77]. It is worth mentioning that protein reorganization is typically discussed in the presence of starch, thus the observations of a lack of homogeneity of the sample should be seen in this context. Recent studies have been published revealing speculations that the protein might have been little involved in the formation of the anisotropic structure of the extrudate, thus the formation of covalent protein bonding to a negligible extent only [2]. Wittek et al. [2] speculated the formation of the anisotropic structure in a soy protein isolate during extrusion was generally related on the formation of insoluble protein clusters around the water dispersed phase, which might be water soluble proteins, including other polymers interacting in the extrusion system, suggesting that insoluble aggregated protein with low solubility and high molecular weight were bounded through isopeptide bonds during the pre-processing of soy protein isolate. The process of extrudate formation was initiated within sections A–B indicated within the arrows in Figure 7.

The protein melting was observed at around the 2–3 last pitches of the screw, yet the previous pitches worked as transportation and precooking systems only. The material inside the barrel, which was left on screw before the A–B area, was unfirm, and bright without noticeable signs of melting formation.

To validate the chosen extrusion procedures for comparison purposes, a soy protein concentrate was extruded as a referenced material. The choice was made due to the material composition and structure similarity to oat protein. Moreover, soy protein concentrates were often preferred choices in extrusion systems utilising a single source of protein [11].

### 3.9. Colour

The colour of the oat and soy extrudates has been presented in Table 3. Expressed as tri-stimulus attributes, the colour of oat extrudates was darker than soy extrudates, despite the relative similarity of the initial colour in the raw materials. A dark colour is generally associated with protein content, and the L value negatively correlates with protein content [78]. Recently, oil’s influence on the lightness during extrusion of soy protein isolates was investigated. An increased oil amount from 0 to 8% reportedly shifted L values from 41.7 to 53.8, respectively [79]. The current study did not reveal a significant colour shift in the a* direction. The colour of extruded samples insignificantly differed comparing them to the raw materials. The literature-reported redness development might be associated with a browning reaction induced by the development of Maillard reactions [80]. Despite the change toward both, the red and blue directions were relatively small and insignificant. The colour change between the raw material and processed material expressed as ΔE averaged at 42.57 ± 5.63 and 62.14 ± 4.56 for soy and oat protein extrudates, respectively. The perceivable colour changes were relatively high and might be analytically described as great [81].

### 3.10. Texture of the Extrudate

The texture profile of the extrudates is represented in Figure 8. The hardness of the oat protein extrudate was higher than that of the soy protein concentrate, although the difference between the samples was insignificant. The differences could be attributed to the mismatching in the composition of the samples, which might influence the texture of the extrudates [82]. It has been reported that the protein content impacted the hardness of the extrudates [83], as well as the oil content [79], which was higher in oat extrudates. The peaked hardness of oat extrudates in some measurements reached 20.9 kg, whereas it is believed the limited acceptable hardness by customers in some categories of products such as snacks should not be higher than 200 N. The variance of the fracturability of the oat extrudate was relatively high. The fracturability expressed through the peak force measurements indicates the brittleness or crunchiness of the products [32]. Such a high variance was apparently related to lower homogeneity of the oat protein extrudate; some fractures have been observed at cross-sections despite the plain surface, which seemed relatively homogenous. The surface of the soy protein extrudate was observably rougher than the oat protein extrudate. The total positive area under the curve, the total work performed during the test, was expressed as toughness; the value was higher in oat protein extrudates, which averaged 35.5 ± 7.51 kg × s, while for soy extrudate its average value was 29.8 ± 4.15 kg × s, though the means of toughness between the samples have not differed significantly.

### 3.11. Scanning Electron Microscope

In this study, a scanning electron microscope was employed to reveal the structure of pure oat protein concentrate, avoiding side influence to extrusion properties from major grain components, such as starch or other admixed components, except the inherent components left after protein extraction, which are believed to have little impact on the extrusion process. In addition, the initial protein material which underwent enzymatic extraction might be considered as a proteinaceous material with no or relatively low impact on its native form. The images displaying the protein microstructure at different magnifications, 20 µm and 200 µm, are presented in Figure 9. The magnification levels and power of 15 kV were chosen empirically as best representing the structure of the extruded oat protein. The pictures of closer magnification demonstrate the surface at cell level, yet magnification at 200 µm reveals the structure overall, showing its homogeneity and orientation. The picture in Figure 9a represents the initial formation of the protein extrudate at point A in Figure 7. The structure was loose, ruptured, with no melting observed. The next picture in Figure 9d shows the beginning of protein concentrate melting before the exit of the extruder (Figure 7, Point B). The sample was partially melted, yet no solid structure was visible. Visible ruptures and aggregation into the anisotropic formations were being observed. Although, the intermediate structure tended to break easily down into smaller formations. The tangential section of the extruded product in Figure 9c,f demonstrates the solid extrudate, which was relatively hard to cut, and changed in colour to pale brown. The longitudinal slice revealed that the material’s texture was relatively oriented towards extrusion direction. The surface was smooth and molten, yet any tendency to form a fibrillar structure was not observed. The breaks inside the structure might be formed by water evaporation, as a result of a lack of cooling. The air release during the extrusion might be a precondition for the ruptured structure formation. The noticeable formation of aggregate-clusters seen in Figure 9c might be related to the collapse of air cells. It was mentioned that the collapse of air bubbles during the extrusion can cause such a crater-like structure formation, especially when protein content in the extrudate is high [76]. The investigated oat protein concentrate contained some amount of fibre, which has not been subjected to the extraction and was concentrated along with protein during processing. The increase in fibre correspondingly increases the cell density wherein the air bubbles are formed. As the brans are formed of mostly insoluble fibre, they underwent very limited changes in solubility during the extrusion [84]. Those work as nucleation agents at bubble formation due to an inability to being wetted. In addition, the increase in bran content simultaneously reduces cross-section volumetric expansion, which in the investigated sample was also insignificant. The published data disclosed increase in longitudinal expansion at the increased bran content [84], yet that specific parameter was not measured in the current research. The cross-section of the extrudate (Figure 9b,e) provided a similar view except the extrudate’s structure orientation, which could be described as an anisotropic.

## 4. Conclusions

The oat protein could be effectively concentrated from oat flour by enzymatic extraction yielding more than 75% of the total protein in oat flour. However, the extracted oat protein bounded a substantial amount of oat lipids, which roughly accounted for about 20% in the protein, restricting direct processing in the wet extrusion process. Treating oat protein concentrates in supercritical fluid systems or ethanol solvent extraction successfully decreased the level of lipids avoiding change of the profile of the amino acids. The molecular size of the protein passed enzymatic extraction corresponded to the reported oat globulin fractions. Oat protein functional characteristics such as solubility and foamability were relatively poor. The obtained protein was further successfully extruded by a single screw wet extrusion system. The observed extrudate could be characterized as the product with relatively high hardness and toughness, the structure of which was anisotropic and highly fractural. The extracted oat protein fraction after defatting could be used in extrusion systems to some extent, or supplementarily.

## Figures and Tables

**Figure 1 foods-12-02333-f001:**
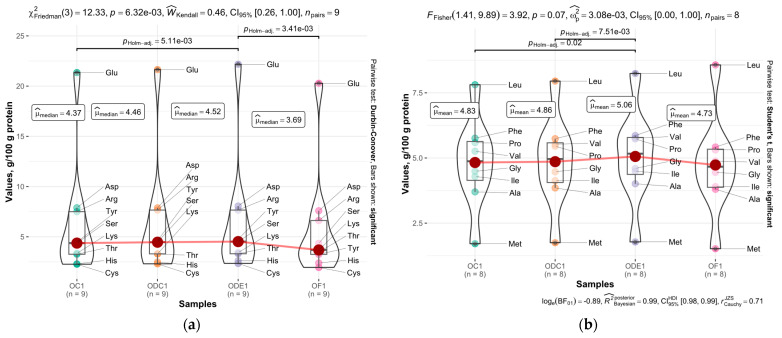
Redistribution of amino acids among the samples: (**a**) polar amino acids in samples OC1, OF1, ODC1 and ODE1; (**b**) non-polar amino acids in samples OC1, OF1, ODC1 and ODE1; only significant differences within the plot are connected by paths to highlight the significance within the group means or medians.

**Figure 2 foods-12-02333-f002:**
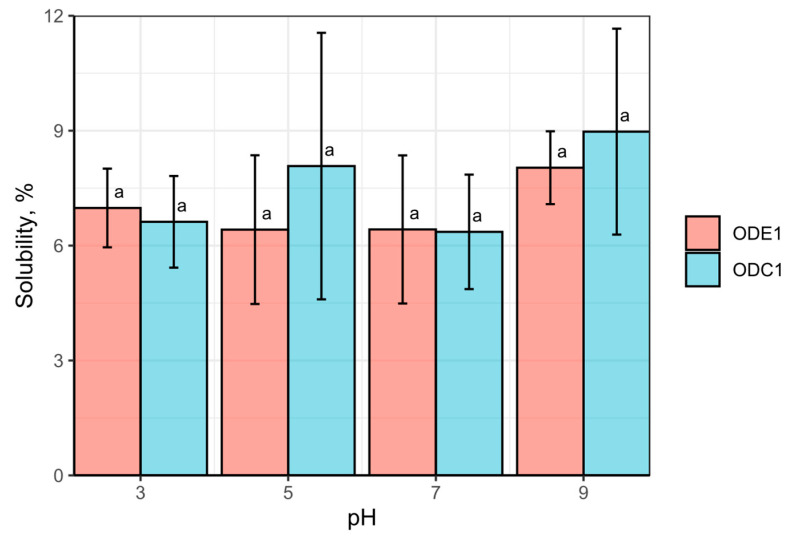
Oat protein solubility in samples ODC1 and ODE1, %. Means sharing common letter do not differ significantly at 5% level of significance.

**Figure 3 foods-12-02333-f003:**
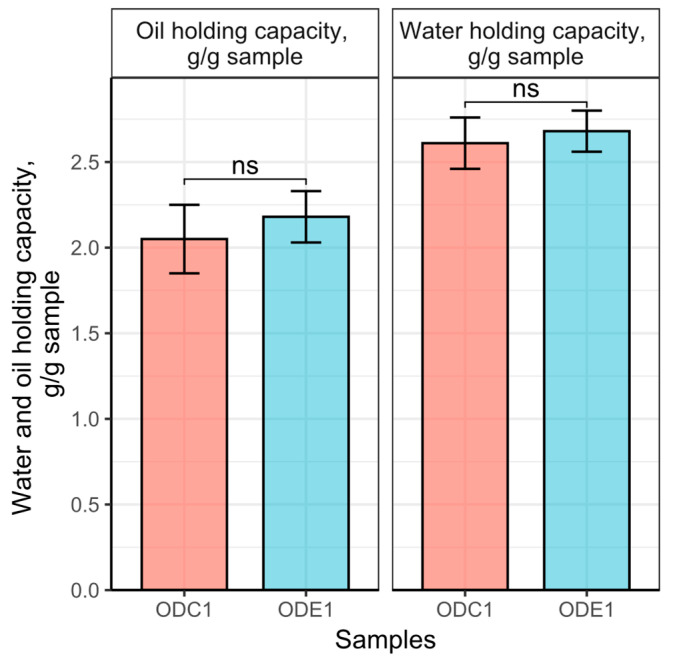
Water and oil holding capacity in samples ODC1 and ODE1, g/g sample. Means represented with “ns” are not significantly different by *t*-test at the 5% level of significance.

**Figure 4 foods-12-02333-f004:**
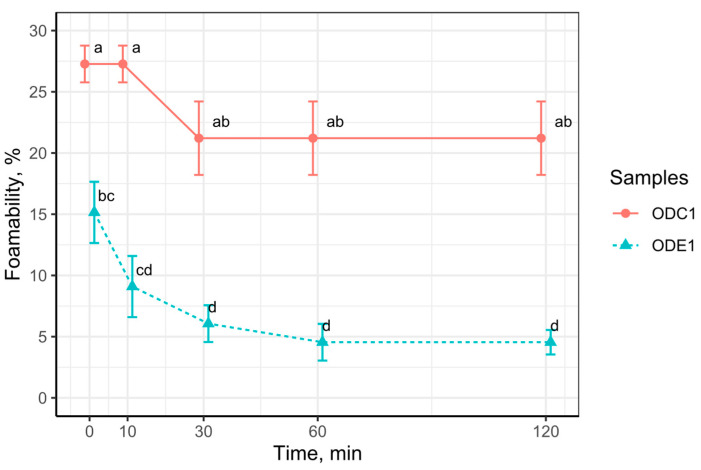
Foaming capacity in samples ODC1 and ODE1, vol%. Means with no letter in common are significantly different (*p* < 0.05).

**Figure 5 foods-12-02333-f005:**
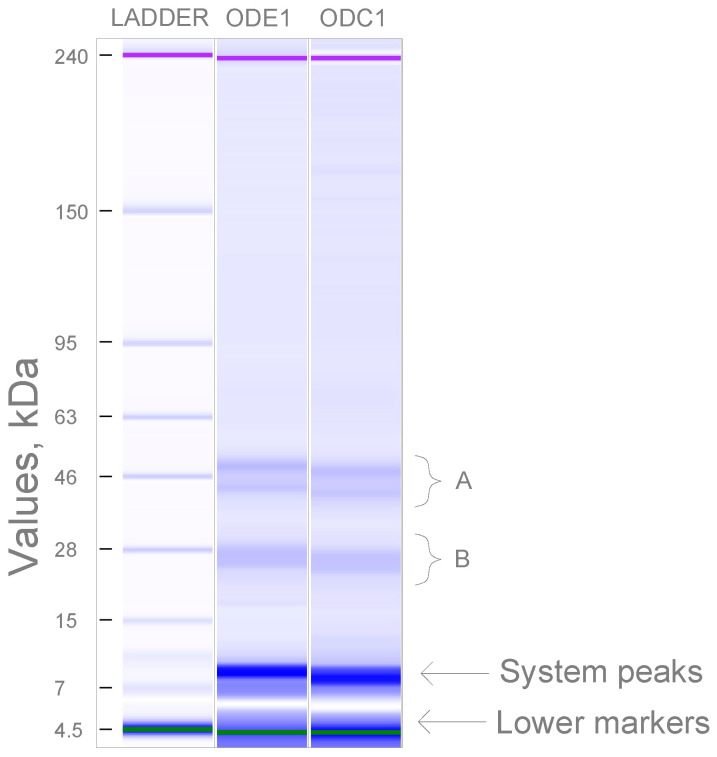
SDS-PAGE image of protein profiles of ODC1 and ODE1. Lower markers and system peaks indicated by arrows did not derive from the analysed samples.

**Figure 6 foods-12-02333-f006:**
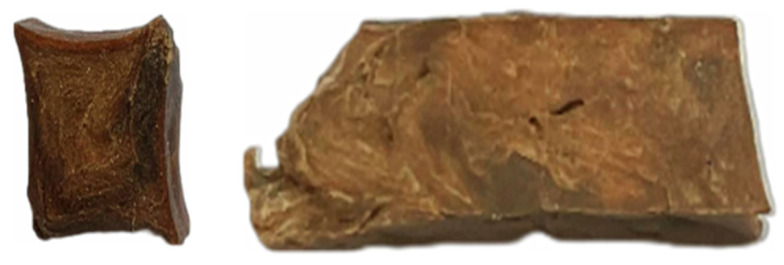
Samples of extrudate, cross and longitudinal section.

**Figure 7 foods-12-02333-f007:**
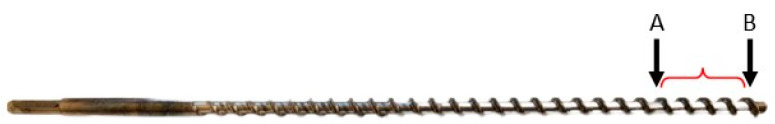
Extrudate formation sections.

**Figure 8 foods-12-02333-f008:**
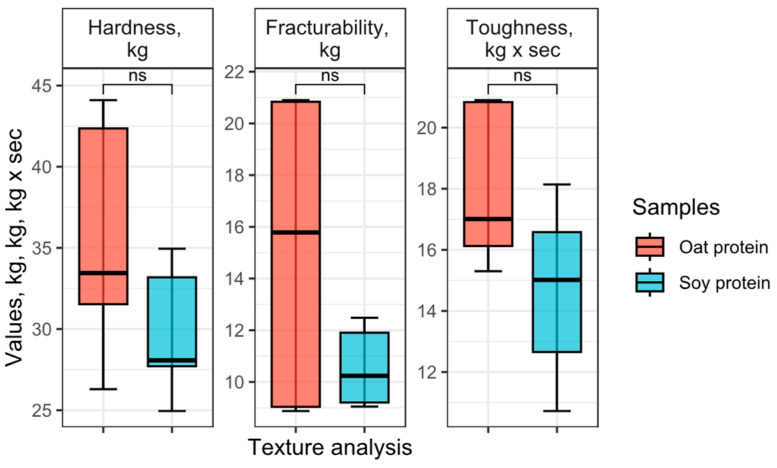
Texture profile of oat and soy protein extrudates, displaying hardness, fracturability and toughness in samples, kg, kg, kg × s. Means represented with “ns” are not significantly different by *t*-test at the 5% level of significance.

**Figure 9 foods-12-02333-f009:**
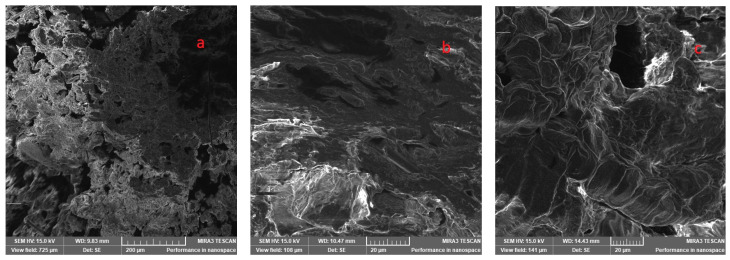
Protein microstructure of oat protein extrudate at different stages of extrudate formation and magnification: (**a**) beginning of the extrusion, longitudinal view 200 µm; (**b**) cross-section 20 µm, (**e**) cross section 200 µm; (**d**) intermediate 200 µm; (**c**) longitudinal view 20 µm, and (**f**) longitudinal view 200 µm.

**Table 1 foods-12-02333-t001:** Chemical composition of the samples. Means within the column not sharing any letter are significantly different at 5% level of significance.

Sample	Protein, DS, wt, %, (N × 6.25)	Crude Oil, DS, wt, %	Fibre, DS, wt, %
Oat Flour	10.44 ± 0.32 c	6.21 ± 0.28 b	0.59 ± 0.21 b
Oat protein concentrate	63.05 ± 1.3 b	20.55 ± 0.4 a	1.34 ± 0.07 a
Defatted by ethanol	78.18 ± 1.93 a	4.88 ± 0.01 c	n.a.
Defatted by CO_2_	77.39 ± 1.58 a	3.48 ± 0.11 d	n.a.

**Table 2 foods-12-02333-t002:** Amino acid composition of analysed protein in initial raw material, OC1, ODE1 and ODC1 and indispensable amino acid requirements in adults regarding FAO recommendations, g/100 g protein. Means followed by the same letter within a row indicate no significant difference among the samples (*p* < 0.05).

Amino Acid	g/100 g in Oat Flour Protein	g/100 g in OC1 Protein	g/100 g in ODE1 Protein	g/100 g in ODC1 Protein	WHO/FAO, Adults, g/100 g Protein **
Ala	3.8 ± 0.17 a	3.7 ± 0.05 a	4.01 ± 0.17 a	3.85 ± 0.06 a	
Arg	6.62 ± 0.29 b	7.5 ± 0.25 a	7.66 ± 0.32 a	7.66 ± 0.08 a	
Asp	7.59 ± 0.24 a	7.89 ± 0.16 a	8.06 ± 0.18 a	7.88 ± 0.3 a	
Cys	1.95 ± 0.05 b	2.27 ± 0.07 a	2.35 ± 0.05 a	2.31 ± 0.07 a	0.6
Phe	5.42 ± 0.27 a	5.76 ± 0.23 a	5.86 ± 0.17 a	5.74 ± 0.16 a	3.8 *
Gly	4.45 ± 0.1 a	4.49 ± 0.19 a	4.6 ± 0.12 a	4.47 ± 0.18 a	
Glu	20.28 ± 0.54 b	21.35 ± 1.01 ab	22.17 ± 0.54 a	21.64 ± 0.38 ab	
His	2.39 ± 0.05 bc	2.3 ± 0.08 c	2.61 ± 0.05 a	2.49 ± 0.06 ab	1.5
Ile	3.9 ± 0.06 b	4.29 ± 0.19 ab	4.49 ± 0.15 a	4.13 ± 0.19 ab	3.0
Leu	8.57 ± 0.11 a	7.81 ± 0.1 b	8.24 ± 0.29 ab	7.95 ± 0.18 b	5.9
Lys	3.69 ± 0.12 a	3.34 ± 0.11 b	3.42 ± 0.12 ab	3.3 ± 0.09 b	4.5
Met	1.52 ± 0.08 b	1.71 ± 0.02 a	1.78 ± 0.04 a	1.75 ± 0.08 a	1.6
Pro	5.31 ± 0.23 a	5.59 ± 0.18 a	5.73 ± 0.17 a	5.45 ± 0.06 a	
Ser	4.34 ± 0.18 a	4.37 ± 0.1 a	4.52 ± 0.15 a	4.46 ± 0.16 a	
Tyr	3.25 ± 0.17 b	4.47 ± 0.14 a	4.75 ± 0.19 a	4.61 ± 0.06 a	
Thr	3.47 ± 0.11 a	3.26 ± 0.16 a	3.31 ± 0.1 a	3.3 ± 0.15 a	2.3
Val	4.88 ± 0.09 c	5.26 ± 0.21 bc	5.76 ± 0.14 a	5.53 ± 0.14 ab	3.9

* Phenylalanine + tyrosine; ** Reprinted with permission from Ref. [50].

**Table 3 foods-12-02333-t003:** CIE lab colour parameters of oat and soy raw materials and extrudates. Means within the column not sharing any letter are significant different by ANOVA test at 5% level of significance.

Sample	a	b	L
Oat raw material	2.6 ± 1.41 a	11.57 ± 5.36 a	83.6 ± 2.47 b
Oat extrudate	0.61 ± 1.56 ab	11.34 ± 5.68 a	21.78 ± 5.27 d
Soy raw material	−0.67 ± 2.38 b	14.19 ± 5.61 a	90.68 ± 2.49 a
Soy extrudate	−0.54 ± 1.55 b	15.48 ± 4.73 a	48.68 ± 3.44 c

## Data Availability

Data is contained within the article.

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
