# Peer review of "Characterisation of the Enzymatically Extracted Oat Protein Concentrate after Defatting and Its Applicability for Wet Extrusion"

_foods, 2023, doi:10.3390/foods12122333_

Round 1

Reviewer 1 Report (Previous Reviewer 3)

Author Response

Good day, 

Please find attached the response. 

Reviewer 2 Report (Previous Reviewer 1)

I suggest recalculating proteins using the correct Nitrogen to Protein conversion factor. 6.25 is very general, and there are more accurate factors available for oats. (Mariotti, François, Daniel Tomé, and Philippe Patureau Mirand. "Converting nitrogen into protein—beyond 6.25 and Jones' factors." Critical reviews in food science and nutrition48.2 (2008): 177-184.)

I am not sure if Figure 1 is the best way to represent the amino acids. I suggest presenting results in the form of a Table.

Figure 3: you should use different symbol for statistics of oil holding capacity and water holding capacity. It seems that you have included all data together. I suggest using small letter (a) for WHC, and capital letter (A) for OHC.

Figure 4: Similarly here, the statistic should be reviewed. You must probably have to do an multiway ANOVA test to see if there is interaction in between the parameters. it does not really makes sense to compare ODC1 at 120 min with ODE1 at 0 min.

L713: I am not sure to understand why the defatting step would impact the protein size. Defatting would not hydrolyze the proteins.

In the version of the manuscript I have downloaded the LaB value are not presented in a table but just as text. I suggest ensuring that the format is correct

Author Response

Good day,

Please find attached the response. 

This manuscript is a resubmission of an earlier submission. The following is a list of the peer review reports and author responses from that submission.

Round 1

Reviewer 1 Report

I do not recommend publication of this paper for the following reasons:

1) While the manuscript entitled "Single Screw Extrusion of Enzymaticlly Extracted Oat Globulins" suggest an emphasis on extrusion, very little information are available on this process. The title should definitely be reviewed as the paper focus on the extraction of globulins from oat using different defatting method and the overall potential for extrusion.

2) The manuscript required intensive English editing for clarification

3) Abstract mentions some conclusions regarding extrusion process that are not discussed in the manuscript (ex. melting and formation of oat protein extrudates were observed at 130 °C)

4) The author should prove that the remaining proteins are globulins using analytical methods

5) Table 2 and Fig. 1 are not necessary in my opinion, they don't bring Information relevant for this study

6) add statistics in every Figures

7)Very little information are provided regarding the extruded product. It would be more relevant to add texture analysis compare to other product

Reviewer 2 Report

Dear Authors,

After reading your paper, I have realized how interesting the topic is. It provides information about isolation of oat protein concentrate which is further defatted by two techniques. In my opinion, the technique which includes ethanol for removing lipids is not adequate. Instead of hexane you can apply some other solvents with similar dielectric constant. Also, it is unusual that both ethanol and CO2 reduced lipids equaly, having in mind that ethanol is adequate for moderately polar compounds. 

In the Abstract, there is an abbrevation OCDE1 that is used only once. Finally, it would be useful to include future perspectives in section Conclusion.

Kind regards

Reviewer 3 Report

The present article titled "single screw extrusion of anymatically extracted oat globulins" in its current form cannot be accepted for publication. It has serious flaws, additional experiments needed (as for example SDS electrophoresis analysis), research not conducted correctly (lack of amino acid description), results presented are chotic, no homogenous groups of statistical analysis presented. Moreover, the proposed title of the manuscript does not fully reflect the analyses contain therein.